



# Seamless climate information for the next months to multiple years: merging of seasonal and decadal predictions, and their comparison to multi-annual predictions

Carlos Delgado-Torres[1], Markus G. Donat[1,2], Núria Pérez-Zanón[1], Verónica Torralba[1], Roberto Bilbao[1], Pierre-Antoine Bretonnière[1], Margarida Samsó-Cabré[1], Albert Soret[1], and Francisco J. Doblas-Reyes[1,2]

[1]Barcelona Supercomputing Center (BSC), Barcelona, Spain
[2]Institució Catalana de Recerca i Estudis Avançats (ICREA), Barcelona, Spain

**Correspondence:** Carlos Delgado-Torres (carlos.delgado@bsc.es)

**Abstract.** Stakeholders across climate-sensitive sectors often require climate information that spans multiple timescales, e.g. from months to several years, to inform planning and decision-making. To satisfy this information request, climate services are typically developed by separately using seasonal predictions for the first few months, and decadal predictions for subsequent years. This shift in information source can introduce inconsistencies. To ensure the information is consistent across forecast time scales, some centres have produced initialised multi-annual predictions, run twice a year and covering 2-3 years ahead, with increased ensemble sizes. An alternative methodology to provide coherent climate information across timescales involves temporal merging, where seamless predictions are created by postprocessing seasonal and decadal forecasts in combination. One approach selects members from large ensembles of decadal predictions or climate projections that closely align with seasonal predictions or past observations, transferring short-term predictability into longer timescales.

This study evaluates the skill of seamless forecasts using different constraints (e.g. variables, regions, temporal aggregations), and compares them with initialised multi-annual predictions. The analysis focuses on predictions of the Niño3.4 index and spatial fields of surface temperature, precipitation, and sea level pressure for the first three forecast years. Results show that while initialised multi-annual predictions achieve the highest overall skill, temporally merged forecasts offer a computationally efficient alternative that still performs well and can be produced regularly as monthly updates of observations or seasonal predictions become available. Besides, both sets of predictions outperform the unconstrained ensembles of decadal predictions and climate projections over large regions. During the period where the seasonal predictions and seamless predictions overlap, their skill is comparable. These findings illustrate the potential of temporal merging as a cost-effective strategy for extending climate information across timescales and enhancing coherence for operational climate services provision.

## 1 Introduction

Climate predictions on seasonal to decadal timescales are increasingly important for decision-making in climate-sensitive sectors such as agriculture, water management, energy and disaster risk reduction (e.g. Torralba et al., 2017; Soares et al., 2018; Paxian et al., 2019; Soret et al., 2019; Turco et al., 2019; Solaraju-Murali et al., 2021; Pérez-Zanón et al., 2024; Delgado-



Torres et al., 2025). These predictions help stakeholders anticipate climate variability and change, enhancing preparation for its impacts. However, user needs often span multiple time horizons (e.g. ranging from months to several years; Merryfield et al., 2020), which requires climate information that is not only skillful but also mutually consistent across these timescales (Kushnir et al., 2019).

Traditionally, seasonal and decadal predictions have been developed and issued separately, often using different forecast systems, ensemble sizes, and initialisation strategies (Goddard et al., 2012; Merryfield et al., 2020). Seasonal forecasts typically focus on the first few months and seasons, and are updated monthly (Weisheimer and Palmer, 2014; Johnson et al., 2019). In contrast, decadal predictions aim to capture longer-term variability and trends over several years, and are produced once per year (Doblas-Reyes et al., 2013; Smith et al., 2019; Delgado-Torres et al., 2022; Hermanson et al., 2022). The discontinuity in the source and structure of the predictions can result in inconsistencies when they are combined to inform medium- to long-term planning.

To address this gap, some forecast centres have produced initialised multi-annual predictions (also referred to as extended seasonal predictions) as part of the EU-funded Horizon Europe ASPECT project (https://www.aspect-project.eu/). This exercise aims to provide seamless information for 2-3 years ahead. These forecasts benefit from larger ensembles and more frequent updates (twice a year) compared to decadal predictions, but are computationally expensive to produce and store.

An alternative and more cost-effective approach is temporal merging, which attempts to build seamless multi-year forecasts by post-processing predictions for different timescales (e.g. seasonal and decadal forecasts) in combination. One strategy is the constraining approach, which selects members from large ensembles of decadal predictions or climate projections that closely align with either recent observations or seasonal forecasts for the next months (Befort et al., 2020; Mahmood et al., 2021, 2022). This method transfers short-term predictability to longer timescales without requiring new multi-annual or decadal predictions, and can be updated whenever new observations or seasonal forecasts are available, offering a computationally cheaper solution.

Previous studies have shown the potential of combining climate predictions at different timescales. For instance, Dirmeyer and Ford (2020) developed a weighting function approach to construct seamless weather-to-subseasonal predictions, and Wetterhall and Giuseppe (2018) generated seamless subseasonal-to-seasonal predictions for hydrological variables. At longer timescales, Befort et al. (2020) showed that constraining projections based on their agreement with decadal predictions improves the surface temperature skill over the North Atlantic Subpolar Gyre region. Mahmood et al. (2021) applied a similar methodology but considered the similarity of SST anomaly patterns for the member selection, finding that regional information can be improved over several parts of the world. Befort et al. (2022) presented evidence that calibrating both decadal prediction and climate projection ensembles together can reduce the inconsistencies when they are concatenated. Similarly, other studies have also shown the benefit of including recent observations to improve the skill of projections already available (Hegerl et al., 2021). For instance, for seasonal predictions of sea surface temperature, Brajard et al. (2023) showed a skill increase through ensemble weighting. At the multi-decadal timescale, Mahmood et al. (2022) found a skill improvement for surface temperature and sea level pressure by selecting those climate projection members based on their agreement with previous observations, and Luca et al. (2023) found an increase in skill for hot, cold and dry extremes using the same approach. In addition, given the mul-



tiple options to decide which members to select, other works have focused on understanding how to best apply the constraints to climate projections (Cos et al., 2024; Donat et al., 2024).

Current research efforts are also devoted to creating seamless information at seasonal and multi-annual timescales. For instance, Abid et al. (2025) combined seasonal and multi-annual predictions by pooling together all the ensemble members that are progressively available, thus increasing the ensemble size as the target period of the forecast is approaching. Navarro et al. (2025) developed seamless seasonal to multi-annual predictions by selecting analogues from transient climate simulations. These produced similar skill patterns to state-of-the-art seasonal and decadal prediction systems, and comparable skill to these initialised predictions, in particular for multi-annual forecast of temperature and standardised precipitation index. Solaraju-Murali et al. (under review) showed the benefit of constraining decadal predictions based on their agreement with the global sea surface temperature pattern predicted by seasonal forecasts.

While both initialised multi-annual predictions and temporal merging techniques aim to provide seamless climate information across timescales, many research questions remain open. For instance, it is unclear how these new sources of seamless climate information perform in different regions and for different climate variables, how they compare to unconstrained decadal predictions and long-term climate projections, and which is the optimal methodology to select the best performing ensemble members. Additionally, the degree to which temporal merging can reach the skill of seasonal predictions in the overlapping period is not known.

This study aims to evaluate the skill of constrained seasonal-to-decadal predictions, and compare them with initialised multi-annual predictions. The analysis focuses on predictions of El Niño-Southern Oscillation (ENSO; a coupled ocean-atmosphere phenomenon in the tropical Pacific that influences weather and climate patterns worldwide, thus impacting sectors such as agriculture, water management, health and renewable energy; McPhaden et al., 2006; Merryfield et al., 2020), and spatial fields of surface temperature, precipitation and sea level pressure over the first three forecast years. We assess skill using different constraints (e.g., variables, regions, temporal aggregation used to select the best ensemble members) and benchmark such skill against the unconstrained decadal predictions and long-term projections. Finally, we also show examples of seamless forecasts to highlight their added value not only in terms of skill improvements, but also in terms of consistency across timescales.

## 2 Data

Climate predictions and projections from forecast systems operating on different timescales are used in this study. For seasonal predictions (SP), we use forecast months 1-6 of the May and November initialisations issued from 1981 to 2014 with the European Centre for Medium-Range Weather Forecasts (ECMWF) fifth generation seasonal forecast system (SEAS5; Johnson et al., 2019), consisting of 25 ensemble members.

For multi-annual predictions (MP), we use forecast months 1-24 from the May and November initialisations produced from 1981 to 2014 with 4 forecast systems (yielding a total of 70 ensemble members; Table S1).

The decadal predictions (DP) are part of the Decadal Climate Prediction Project Component A (DCPP-A; Boer et al., 2016) of the Coupled Model Intercomparison Project Phase 6 (CMIP6; Eyring et al., 2016). Although more recent decadal predictions



are available for some systems (Hermanson et al., 2022; Delgado-Torres et al., 2025), we limit the analysis to initialisations up to 2013, as this is the last initialisation available for all the decadal forecast systems in CMIP6/DCPP. Decadal forecast systems are initialised towards the end of each year in slightly different months. Thus, the first forecast months are discarded for some systems to align all predictions to start in January (i.e. the first two and three months have been discarded for those systems initialised in November and October, respectively). Therefore, we use 60 forecast months of DP initialised at the end of each year from 1980 to 2013 produced with 17 forecast systems (a total of 197 ensemble members; Table S1).

For climate projections (HIST), we use the CMIP6 historical forcing simulations and scenario SSP2-4.5 (O'Neill et al., 2016) produced with 32 different climate models (resulting in a total of 264 ensemble members, Table S2). The CMIP6 historical experiment provides data until 2014, and then is concatenated with the scenario SSP2-45 for the rest of the period (2015-2018).

The ERA5 reanalysis (Hersbach et al., 2020) is used as the observation-based reference dataset (OBS) to calibrate the predictions, rank the members (described in Methods), and evaluate the forecast quality.

We use monthly means of near-surface air temperature (TAS), sea surface temperature (TOS), precipitation (PR) and sea level pressure (PSL). The North Atlantic Oscillation (NAO) index is computed as the difference between the area-weighted average PSL anomalies of the subtropical Mid-Atlantic and Southern Europe region (90ºW-60ºE, 20ºN-55ºN) and the North Atlantic–Northern Europe region (90ºW-60ºE, 55ºN-90ºN), following (Stephenson et al., 2006). The Niño3.4 index is calculated as the area-weighted average TOS anomalies over the east-central tropical Pacific region (170ºW-120ºW, 5ºS-5ºN), following (Barnston et al., 2019), representative of the ENSO state.

## 3 Methods

Prior to the analysis, all the data have been conservatively interpolated to a 1ºx1º horizontal resolution (Schulzweida, 2023). This horizontal resolution has been chosen as a compromise between the different resolutions of the different datasets used in the analysis. Several post-processing steps are then applied, including anomalies computation, bias-adjustment (correcting both the mean and variance), indices calculation and members selection. In the case of SP, the ensemble mean is post-processed because no member selection is applied to this data type. For the rest of the predictions, the ensemble members are post-processed independently. The post-processing steps have been applied in leave-one-out cross-validation, i.e. using the full period but excluding the observations of the year being post-processed in order to emulate real-time conditions and avoid overestimating the actual skill (Barnston and Dool, 1993; Risbey et al., 2021).

The constraining methods applied in this study are based on those introduced by Mahmood et al. (2021, 2022). For each start year, the members of the DP and HIST ensembles are ranked based on their agreement with either the OBS from previous months or SP for the next months. For the OBS-based ranking, the agreement is estimated with the average of the observations of the previous 1, 1-2, 1-3 and 1-4 months. For example, in the case of ranking in May, the DP members initialised in January are compared to the observations of April, March-April, February-April and January-April. For the SP-based ranking, the agreement is estimated using the forecast months 1, 1-2, 1-3, 1-4, 1-5 and 1-6 of the SP ensemble mean. For instance, the ranking in May of a DP member is produced by comparing such a member and the SP ensemble mean for predictions of





May, May-June, May-July, May-August, May-September and May-October. Figure S1 shows an illustration of the different methods.

The agreement for the ranking has been computed based on spatial fields of TOS and PSL, and on the Niño3.4 and NAO indices. In case of spatial fields of TOS and PSL, the spatial correlation, centered-RMSE and uncentered-RMSE (Wilks, 2011) are estimated over the Global (180ºW-180ºE, 90ºS-90ºN), Global without the poles (NoPolar; 180ºW-180ºE, 60ºS-60ºN), Atlantic and Pacific Oceans (Alt+Pac; 120ºE-0ºE, 25ºS-60ºN), Pacific Ocean (Pac; 140ºE-85ºW, 60ºS-60ºN) and North Atlantic Ocean (NAtl; 80ºW-0ºE, 0ºN-60ºN) regions (Figure S2), as in Mahmood et al. (2021). In the case of Niño3.4 and

NAO, the ranking of the members is based on the mean absolute error, inspired by the NAO-matching methodology proposed by Smith et al. (2020). With all the combinations, there are a total of 128 OBS-based constraints (2 indices x 4 months + 2 variables x 4 months x 3 metrics x 5 regions) and 192 SP-based constraints (2 indices x 6 forecast months x 1 metric + 2 variables x 6 forecast months x 3 metrics x 5 regions). This gives a total of 320 constraint-based methods considered. Once the ranking is performed, the best 30 members according to each constraining method are used to build the constrained ensembles

(Best). Mahmood et al. (2021, 2022) found robustness of the results to the choice of the number of selected members.

    The anomaly correlation coefficient (ACC; Wilks, 2011) is used to evaluate the forecast quality of the predictions. The ACC ranges from -1 to 1. Negative or near-zero values mean no forecast quality, while ACC equal to 1 indicates a perfect forecast. The residual correlation (Smith et al., 2019) is applied to estimate the impact of the constraining methods. The residual correlation measures whether a forecast captures any of the observed variability that is not already captured by a reference

forecast, and it is computed as the ACC between the residuals of a forecast and the observations once the reference forecast's ensemble mean has been linearly regressed out from both the forecast's ensemble mean and observations. For instance, if the residual correlation is positive when computed using the Best ensemble as the forecast and the DP ensemble as the reference forecast, it indicates an added value of the constraining methodology on the forecast quality. The statistical significance of the ACC and residual correlation is assessed using a one-sided and two-sided t-test, respectively, at the 95% confidence level.

The timeseries autocorrelation has been taken into account by using the effective number of degrees of freedom following von Storch and Zwiers (1999).

## 4   Results and Discussion

This section is divided into three subsections: first, we show the skill obtained with the different ensembles (i.e. SP, MP, DP and HIST, as well as the Best ensemble built with the 320 constraining methods) as a function of the forecast month for predictions

of the Niño3.4 index. Then, we focus on the skill for spatial fields of TAS, PR and PSL. Finally, we show examples of seamless forecasts, selecting one Niño and one Niña event. The focus is set on assessing the benefits of applying constraints to produce seamless forecasts, as well as on identifying which constraining method provides the highest skill (e.g. which variable and which region are best to select the best members).





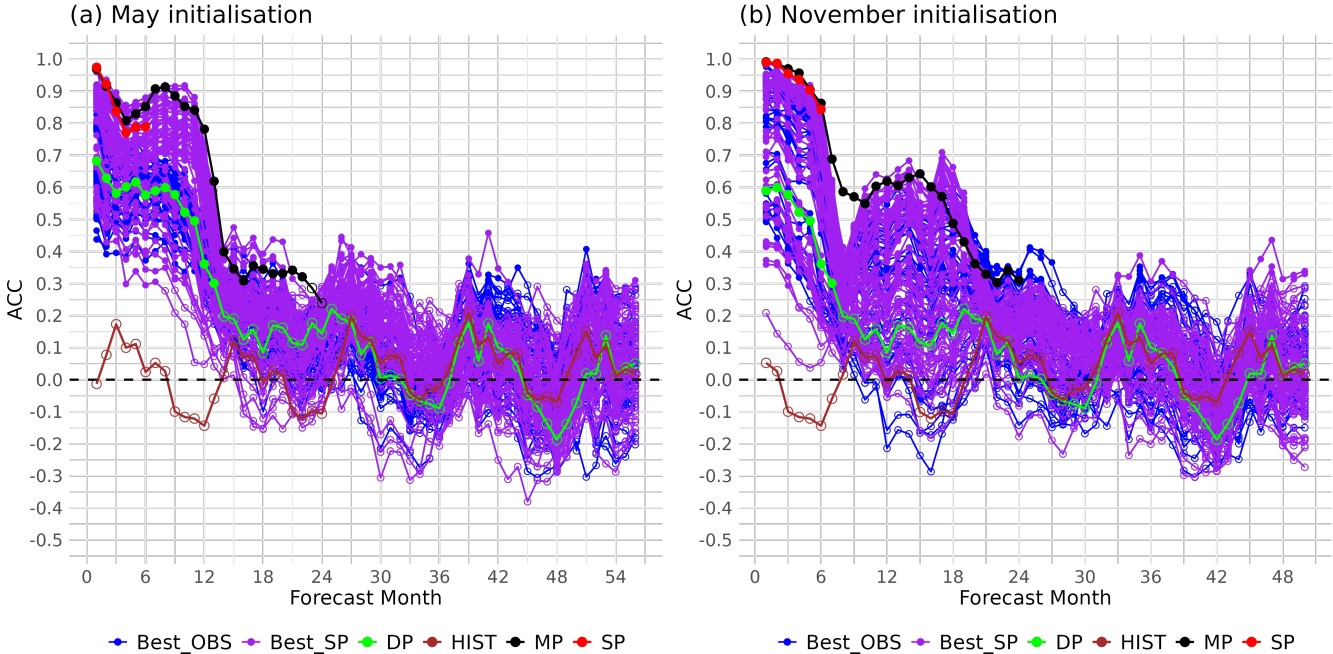

**Figure 1.** Forecast quality for the Niño3.4 index. ACC as a function of the forecast month for predictions issued in May (left) and November (right). The forecast quality is shown for SP (red), MP (black), DP (green), Best_OBS (blue), Best_SP (purple), and HIST (brown). The ERA5 reanalysis has been used as the reference dataset. Filled dots indicate statistically significant ACC using a one-sided t-test at the 95% confidence level accounting for timeseries autocorrelation.

## 4.1 Assessment of the Niño3.4 forecast

The skill as a function of the forecast month for predictions of the Niño3.4 index issued in May and November is shown in Figure 1. The SP forecast quality is high and statistically significant during the six forecast months for both initialisation months, with ACC values close to 1 during the first forecast months, and progressively decreasing to values close to 0.8 for the forecast month 6. The skill of the MP is also high and significant during the first forecast months, and is very similar to that of SP during the overlapping period (first six months). However, a strong skill decrease can be seen when the forecasts approach April-May (approximately forecast month 12 and 6, respectively, for the May and November initialisations), consistent with the spring predictability barrier (Duan and Wei, 2013; Ehsan et al., 2024). Still, the MP skill is significant up to forecast month 22 for the May initialisation, and forecast month 24 for the November initialisation. In particular, the MP skill is still 0.5 for the forecast month 18 (Figure 1b), which indicates that the ENSO state can be predicted with reasonable skill for the first two winters after the November initialisation.

The DP multi-model also shows statistically significant skill during the initial forecast months up to boreal spring. However, the skill is lower than for both SP and MP. For instance, the skill for forecast month 1 is close to 0.7 and 0.6, respectively, for predictions issued in May (Figure 1a) and November (Figure 1b). This reduced performance is primarily due to the inherent



delay in the availability of DP: forecast month 1 in DP does not actually correspond to the first calendar month after initialisation. Instead, DP issued in May are based on predictions initialised 5 to 7 months earlier (in October, November or January depending on the forecast system; see Table S1). Likewise, those DP issued in November are based on forecasts initialised 10 to 12 months prior. This lag reflects the operational realities of producing decadal predictions, which involve generating initial conditions, running complex forecast systems, post-processing outputs, and assembling multi-model data products. As a result, the multi-model DP initialised around the end of the previous year are typically only becoming available around May, which corresponds to lead month 7 for a forecast initialised in the previous November (green line in Figure 1a). Similarly, the DP issued in November would rely on forecasts initialised the prior year (green line in Figure 1b). This explains the skill difference between, for example, the DP and MP ensembles during the first forecast months. It should be noted that the MP would also have a delay between their production and availability. However, we prefer to use MP from their first forecast month because (1) all MP systems are initialised in November and (2) we use them in this study for comparison to the constrained predictions. Thus, we show the potential skill that the MP would have if they were available immediately after initialisation.

For reference and for a fair comparison with MP, the skill of the first forecast months (from the first January, as some of the models are initialised in the month) of DP without taking into account the delay until they become available can be found in Figure S3 (i.e. MP being initialised in November, and DP towards). The skill of HIST, which is not initialised and thus not in phase with the observed internal climate variability, is low and not significant (ranging between -0.2 and 0.2), with some seasonality likely linked to the annual cycle of ENSO activity (Ehsan et al., 2024).

The skill of the constrained ensembles varies considerably across the 320 different OBS-based and SP-based methods. For instance, the constrained ensemble with the lowest skill for the forecast month 1 shows a correlation close to 0.4 for the May initialisation (0.2 for the November initialisation). In contrast, the highest-performing constrained ensembles exhibit skill levels comparable to those of the MP ensemble. To our knowledge, this is the first time it is reported that the skill of state-of-the-art initialised multi-annual predictions represents an upper bound that cannot be surpassed by any of the tested seasonal-to-decadal constraining approaches for the Niño3.4 index out to two years. This suggests that, although constraining methods can enhance the skill of the forecasts, they cannot outperform forecasts from a state-of-the-art initialised system. A possible explanation lies in the limited number of climate states available for selection within the DP and HIST ensembles when applying the constraining methodology. For longer forecast time averages (forecast years 1-5 and 1-10), Mahmood et al. (2022) and Donat et al. (2024) showed that the constrained HIST ensembles can provide regionally higher skill than the initialised DP against which they were constrained.

However, even if the MP skill is higher than that of the constraining method, such skill is comparable and, given the difference in computational resources to create these two sources of seamless climate information (MP being much more computationally expensive than applying constraints to existing simulations), the constraining approach seems a cost-effective alternative. In addition, the MP would not be immediately available right after the initialisation, so some delay and its associated skill decrease would be expected in real-time production. Furthermore, the constrained ensemble could be produced at any time of the year, once new observations or seasonal predictions become available, without the need of running the longer-term (MP or DP) forecast systems more than once per year.





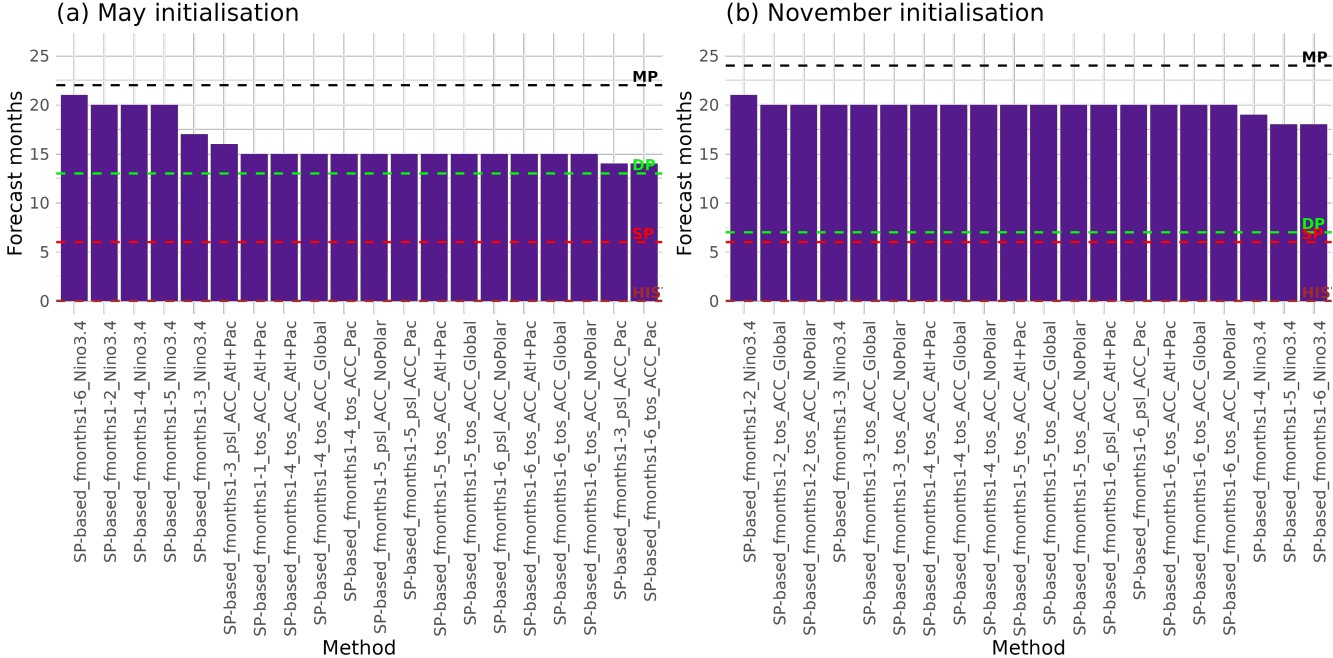

**Figure 2.** Top methods to constrain predictions for the Niño3.4 index. The ranking of the different methods is based on the number of forecast months showing statistically significant ACC consecutively until the first forecast month that is not significant for predictions issued in May (left) and November (right). Only the best 20 methods are shown (see Methods for the definition of the different methods). The same score is shown for SP (red), MP (black), DP (green) and HIST (brown). The ERA5 reanalysis has been used as the reference dataset. The ACC statistical significance has been computed using a one-sided t-test at the 95% confidence level accounting for timeseries autocorrelation.

In order to identify which is the best constraining method to create a seamless forecast for the Niño3.4 index, we have given a score to each method. The score is defined as the number of consecutive forecast months that show significant skill until the first forecast month for which the skill is not significant (i.e. the last filled dot before the first empty dot in Figure 1). For comparison, the same score is calculated for the SP, MP, DP and HIST ensembles. The top-ranked method for seamless predictions of the Niño3.4 index issued in May is the one based on the SP of the Niño3.4 index for the forecast months 1-6, providing significant skill for the first 21 forecast months (Figure 2a). This method is followed by similar constraints, but using the SP for the forecast months 1-2, 1-4 and 1-5 (20 significant forecast months). After these methods, the scores decrease to ~15 significant forecast months for constraints based on the agreement of spatial patterns of TOS and PSL, being all of them SP-based methods. All the top-ranked methods outperform the score obtained with the unconstrained DP (score of 13 forecast months) and HIST (which shows no significant skill for any forecast months). However, no method reaches the 22 forecast months of the MP.

Smaller differences among the top-ranked methods are found for the November initialisation (Figure 2b). In this case, the best method shows significant skill until forecast month 21 (corresponding to constraining based on SP of the Niño3.4 index



for the forecast months 1-2). The following methods are mostly based on spatial fields of TOS, and show a score of 20 forecast months. In this case, DP are statistically significant until forecast month 7, and HIST gets a score of 0. Please note that, in case of the DP, most of the forecast systems are initialised one year before, which explains why the quality is relatively low when the actual forecast is used one year after initialisation. It is interesting to see that, for both initialisation months, at least the top 220 twenty-ranked methods are based on SP, while no method based on OBS appears in Figure 2.

The results of the constrained ensembles shown in Figures 1 and 2 correspond to the best 30 members selected from both the DP and HIST ensembles. We have also tested the sensitivity of the results when the members are selected only from either the DP ensemble or HIST ensemble (Figures S4 and S5). In the case of selecting members only from the DP ensemble the skill of the constrained ensembles is higher than DP for most of the constraining methods. On the other hand, when the best 225 members are selected only from the HIST ensemble, the skill of the constrained ensembles varies more and tends to be lower, with a large number of constraining methods providing lower skill than the unconstrained DP ensemble (and in some case, even lower than the unconstrained HIST ensemble).

The identification of the best constraining method has been carried out for predictions of the Niño3.4 index. However, such a best method may be different if other indices (e.g. the NAO index) or variables over specific locations (e.g. precipitation over 230 Europe) are considered. Thus, a specific evaluation of this methodology should be applied to identify the optimal selection approach for issuing seamless forecasts for particular regions and variables.

## 4.2   Forecast assessment for spatial fields of TAS, PR and PSL

In the previous section, the ensemble selection is conditioned on the quality of Niño3.4 index predictions. Therefore, those methods are expected to perform well over the Niño3.4 region and other regions teleconnected with ENSO. However, the 235 performance of those methods may be suboptimal elsewhere. Therefore, we also identify the methods that provide the highest overall skill for global predictions of TAS, PR and PSL.

First, we consider the constraining method based on SP of the Niño3.4 index for the forecast months 1-6 (identified as the best method in Figure 2a, i.e. the Best ensemble containing the top-ranked 30 members), selecting from both the DP and HIST ensembles. The number of members per model and start date is shown in Figures S6 and S7 for May and November 240 initialisations, respectively. For the May initialisation, the constrained Best ensemble shows significantly positive skill for TAS over large parts of the globe for all the forecast periods analysed (Figures 3a,e,i,m). Specifically, this Best ensemble exhibits significant skill over 75.3% of the globe for the forecast months 1-6, and 76.1%, 64.5% and 60.8% of the global regions for the forecast years 1, 2 and 3, respectively.

During their overlapping period, the Best and SP ensembles show broadly similar skill for TAS during the forecast months 245 1-6 (Figure 3b). Notably, Best is significantly better than SP over 8% of the region, while SP is better than Best over 2.1%, as shown with the positive and negative residual correlations, respectively. This relatively small difference indicates that the Best ensemble can be used to issue a seamless forecast directly from its production date (in this case, May), without the need to initially use SP and then switch to Best. However, it is important to note that SP is still required to constrain the DP and/or







**Figure 3.** Forecast quality for the near-surface air temperature issued in May. ACC obtained with the constrained Best ensemble (first column). Residual ACC obtained with the constrained Best ensemble using SP or MP as the reference forecast (second column), the unconstrained DP as the reference forecast (third column), and unconstrained HIST as the reference forecast (fourth column). The different columns correspond to different forecast periods. The Best ensemble has been built with the constraining method based on SP of Niño3.4 for the forecast months 1-6 selecting from both the DP and HIST ensembles. Dots indicate statistically significant ACC and Residual ACC values at the 95% confidence level using a one-sided (two-sided) t-test for ACC (Residual ACC) accounting for timeseries autocorrelation.

HIST ensembles to generate the Best ensemble. Similar results are found when comparing Best against MP for the forecast

250    years 1 and 2 (Figures 3f,j, respectively).

The comparison of the Best ensemble against the unconstrained DP and HIST ensemble reveals a significant added value of the constraining approach, particularly in tropical regions (Figures 3c,d,g,h,k,l,n,o). For example, the constrained ensemble is significantly better (i.e. significant positive residual correlation) than DP over 29.8% and 12.2% of the region for the forecast years 1 and 2, respectively, while it has significant negative residual correlations only over 1.1% and 1.3% (Figures 3g,k).





Regarding the comparison of Best and HIST, the constrained ensemble has significant positive residual skill than HIST over the 45.1% and 22.6% of the region for the forecast years 1 and 2, respectively (Figures 3h,l). These results are based on a single constraining method; other methods may yield higher skill in specific regions, pointing to the need to tailor the constraining strategy to the region of interest.

The skill for PR is lower than for TAS, in line with previous studies (e.g. Smith et al., 2019; Delgado-Torres et al.,
2022, 2023). Nevertheless, the constrained ensemble shows significant skill over 26.6% of the region for the forecast months 1-6, and 31.4%, 15% and 10.1% for the forecast year 1, 2 and 3, respectively (Figure 4a,e,i,m). The added value of the constraining approach for predictions of PR is also lower than for TAS. Still, the fraction of the global region showing significantly positive residual correlations is also higher than the fraction of negative residual correlation. For example, Best outperforms SP over 5.5% of the area, while Best is worse than SP over 2% (Figure 4b). Similarly, Best is better than MP over the 6% and
4.8% for the forecast years 1 and 2, while it is worse over 1.8% and 1.5% (Figures 4f,j). Regarding the comparison of Best against DP and HIST, the fraction of significant area is also higher for all the forecast periods analysed, particularly for the shorter timescales. For example, Best is better than DP and HIST over 14.5% and 23.8% of the region for the forecast year 1, whereas it is worse over 0.9% and 0.6% of the region (Figures 4g,h).

For PSL, the Best ensemble shows significant skill over 53.8%, 56.7%, 25.2% and 22.9% of the globe for the forecast months
1-6, year 1, 2 and 3, respectively (Figures 5a,e,i,m). The comparison of Best against the different reference forecasts shows a significant added value of the constraining approach for all the forecast periods evaluated: 10%, 4.1% and 7.7% of the region shows significant improvements for the forecast months 1-6 against SP (Figure 5b), and forecast years 1 and 2 against MP (Figures 5f,j), while Best is worse than SP and MP over 0.7%, 1.2% and 0.7%, respectively. The constrained ensemble is also significantly better than the unconstrained DP and HIST over large regions. For instance, 33.4% and 14.8% of the globe shows
significantly positive residual correlation when comparing Best against DP for the forecast years 1 and 2, while only showing 0.8% and 0.4% of significantly negative residual correlation (Figures 5g,k). Similarly, the constraining approach shows a significant added value when compared to HIST. For instance, 50.6% and 16.1% of the globe show significantly positive residual correlation for the forecast years 1 and 2, respectively, while only 0.5% and 0.4% of the region show significantly negative residual correlation.

Figures 3-5 show the ACC of Best, and its comparison with different reference forecasts (SP, MP, DP and HIST) through the residual ACC, but the actual ACC values of such reference forecasts are presented in Figure S8-S10. In addition, the results for the November initialisation are also provided in the Supplementary Material (Figures S11-S16).

As for the Niño3.4 index, we investigate the sensitivity of the constrained forecast quality to the different constraining choices (e.g. variable, region, metric), and identify the methods providing the highest global skill for TAS, PR and PSL. For
each of the 320 constraining methods (see Methods), we calculate the fraction of the global region with significant ACC, and compare subsets of these combinations to assess sensitivity (Figure 6). For reference, we also include the performance of the unconstrained ensembles (i.e. SP, MP, DP and HIST).

We begin by assessing the sensitivity to the choice of ensemble. Overall, selecting members from either DP or DP+HIST results in similarly high skill. However, the DP+HIST ensemble displays a broader range of outcomes. Therefore, selecting



**Figure 4.** Forecast quality for precipitation issued in May. Same as Figure 3, but for precipitation.

only from DP appears to be the preferable approach, as it tends to produce distributions of significant areas that are more tightly constrained toward higher values. However, given the lower availability of DP in real-time than for the historical period (Delgado-Torres et al., 2025), selecting from both DP+HIST seems a reasonable trade-off between skill and operational feasibility. On the other hand, selecting only from HIST results in the overall lowest skill, particularly for the forecast months 1-6 and forecast year 1 (Figures 6a,b,c). SP and MP outperform the constraining approaches, particularly at shorter timescales.

The sensitivity of the skill when constraining based on past OBS or future SP shows that, in general, the larger fraction of significant skill is achieved for the SP-based constraints (Figures 6d,e,f). Again, the sensitivity is higher during the shorter forecast periods. Comparing constraints based on spatial patterns of TOS and PSL, results are very similar (Figures 6g,h,i), though slightly higher significance can be obtained when using TOS for predictions of TAS for the forecast months 1-6 (Figure 6g), and when using PSL for predictions of PR and PSL for the forecast year 1 (Figures 6h,i).





**Figure 5.** Forecast quality for precipitation issued in May. Same as Figure 3, but for sea level pressure.

There are no clear differences in skill significance across different constraining regions (Figures 6j,k,l), likely because different regions provide predictability over different parts of the globe (e.g. North Atlantic constraints may benefit teleconnected regions). Similarly, the choice of metric for selecting members does not substantially affect the fraction of significant area (Figures 6m,n,o), though selecting members based on absolute error with respect to indices such as the NAO or Niño3.4 yields noticeably lower skill.

Consistent results are found for the November initialisation (Figure S17). The low sensitivity found for the different constraining choices may be due to the global spatial averaging. Therefore, to identify the optimal constraining method for applications at a specific location, the analysis should be repeated, as the results might vary depending on regional characteristics.





**Figure 6.** Sensitivity to constraining methods and parameters for predictions issued in May. Fraction of global area showing statistically significant ACC at the 95% confidence level using a one-sided t-test accounting for timeseries autocorrelation. The sensitivity analysis has been carried out for the selectable ensemble (DP, HIST or DP+HIST; first row), the selection type (OBS-based or SP-based; second row), the constraining variable (TOS or PSL; third row), the constraining region (Niño3.4, NAO, Global, NoPolar, Atl+Pac, Pac, NAtl; fourth row), and the constraining metric (mean absolute error with respect to the NAO or Niño3.4, and spatial ACC, spatial centered-RMSE or spatial uncentered-RMSE with respect to TOS or PSL; fifth row). See Methods for a full description of all the constraining methods tested. The results are shown for TAS (first column), PR (second column) and PSL (third column).





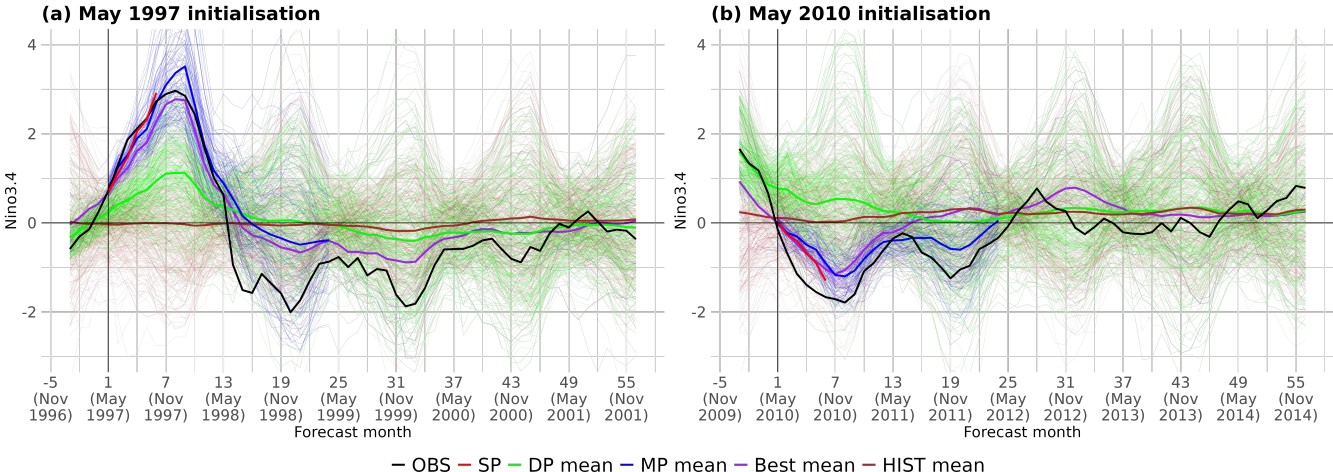

**Figure 7.** Seamless forecast of the Niño3.4 index produced in May 1997 (a) and May 2010 (b). Time series Niño3.4 index of observations (black), SP produced in May (red), MP produced in May (blue), DP produced at the end of the previous year (green), Best constrained in May (purple) and HIST (brown). The Best ensemble has been built with the constraining method based on SP of Niño3.4 for the forecast months 1-6 selecting from both the DP and HIST ensembles. The thin lines correspond to the ensemble members, while the thick lines correspond to the ensemble means.

### 4.3   Examples of seamless forecasts for the 1997-98 el Niño and 2010-11 la Niña events

Finally, we present some examples of seamless forecasts to illustrate not only the benefit of the constraining approaches in terms

of improved skill, but also their added value in enhancing forecast consistency across timescales. We focus on the forecasts produced in May 1997 and May 2010, which successfully captured the following El Niño and La Niña events, respectively. These two events were selected because they are among the strongest and most impactful ENSO events on record, making them ideal test cases to assess the ability of the methodology to capture key climate signals. Figure 7 shows the observed Niño3.4 index and the different forecasts (i.e. SP, MP, DP, HIST and Best).

The case of May 1997 (Figure 7a) is particularly relevant as the 1997-1998 Niño event was exceptionally strong (Trenbeth et al., 2002), with widespread global impacts (McPhaden, 1999; Rojas et al., 2014). The SP correctly captured the evolution of OBS during the forecast months 1-6. Similarly, the MP accurately predicted the development and peak of the El Niño event, with the maximum occurring around December 1997, when the observed Niño3.4 index reached around +3ºC. In contrast, the DP initialised in late 1996 indicated a positive ENSO phase but with a much lower amplitude (around +1.2ºC). As expected,

the uninitialised HIST ensemble showed no clear ENSO signal and failed to capture the event.

    The constrained Best ensemble based on the Niño3.4 for the forecast months 1-6, however, successfully predicted the strong El Niño event, showing performance comparable to that of MP. This example highlights that both MP and the constrained Best ensemble can be used to generate seamless forecasts that provide consistent and skilful climate information. Conversely, using SP for the first six months and then switching to DP from month seven introduces a large inconsistency: a sudden difference of





1.7°C in the forecasted index (+2.9°C from SP and +1.2°C from DP). This discontinuity is avoided by the seamless approaches that not only increase the skill but also improve the temporal coherence of climate predictions across lead times.

A similar behaviour is observed for the May 2010 forecasts (Figure 7b), which preceded two consecutive La Niña events during 2010-2012, also being one of the most important Niña events on record (Boening et al., 2012; Feng et al., 2013), with severe and unprecedented impacts (Hoyos et al., 2013; Vargas et al., 2018). In this case, SP also captured the ENSO phase,

although with lower accuracy than seen for the 1997-1998 case. By contrast, both DP and HIST failed to predict the correct ENSO phase: HIST showed no signal, and DP indicated a weak warm anomaly that gradually returned to neutral conditions. However, both the initialised MP and the constrained Best ensembles captured the ENSO phase. The MP also predicted the second Niña event (2011-2012), although its intensity was underestimated.

## 5   Summary and Conclusions

This study presents and evaluates a methodology for producing seamless climate forecasts from 1 month to 2-3 years ahead by combining predictions across different timescales. The proposed approach, which constrains large ensembles of decadal predictions or climate projections based on observations of the previous months or seasonal forecasts, aims to reduce inconsistencies in the forecasts when switching from seasonal to decadal sources of climate information, and to enable the generation of temporally coherent climate information that can be applied in real-time conditions and tailored to specific user needs.

Constraining forecasts based on their agreement with seasonal predictions leads to higher overall skill than constraining based on previous observations, particularly for the Niño3.4 index. We find that the skill of initialised multi-annual predictions sets an upper bound that is reached by some constraining approaches, though not outperformed by any. However, these multi-annual predictions are typically not produced in real time and, due to their high computational cost, are usually only issued a few times per year by most forecasting centres. In contrast, the temporal merging approach provides a cost-effective alternative

to emulate their performance using operationally available data. It enables the transfer of short-term predictability to longer-term forecasts and can be updated much more frequently (potentially every month) when new seasonal forecasts or observations become available.

We demonstrate that selecting members only from the (also initialised) decadal prediction ensemble generally provides higher skill than selecting from a combination of decadal predictions and (uninitialised) climate projections, or from climate

projections alone. However, in real-time forecasting, the availability of decadal predictions is lower, requiring the inclusion of HIST members to ensure a large ensemble size. This introduces a trade-off between forecast skill and operational feasibility.

The Niño3.4 index benefits from the constraining methodology, showing significant skill up to approximately 20 months ahead. This highlights the value of seamless forecasting systems in capturing the ENSO variability, which have widespread societal impacts. By examining specific case studies (1997-1998 El Niño and 2010-2012 La Niña), we show that the constrained

ensemble not only reproduces these events with accuracy, but also that the methodology avoids discontinuities between seasonal and decadal forecasts. In contrast, using SP followed by DP forecasts results in inconsistencies, such as jumps in predicted values during the transition between timescales. Moreover, during their overlapping forecast period, constrained seamless



predictions are as skillful as seasonal forecasts, which suggests that the constrained ensemble can be used from the beginning of the forecast period without losing skill.

Furthermore, we find that, in terms of fraction of the global area showing statistically significant skill, the different seamless forecast methodologies used in this study are robust across a wide range of tested configurations, including different constraints, variables, regions, and metrics. While sensitivity to these factors exists, especially for short lead times, the general result is that forecast quality is preserved or improved when using constrained ensembles. The constraining approach can be extended to user-specific indicators or climate extremes, offering a pathway to operational, consistent, and application-relevant climate

services at seasonal-to-decadal timescales.

    In summary, our results demonstrate that seamless forecasts based on ensemble selection offer a computationally efficient and skillful solution for delivering climate information from seasonal to decadal timescales. By selecting members from large ensembles according to their agreement with recent observations or updated seasonal forecasts, this methodology enhances temporal consistency and prediction skill over large areas of the world. The methodology is particularly valuable in real-time

contexts where multi-annual forecasts are not routinely available, and it can be adapted to deliver tailored information for specific sectors, indicators, or extreme indices. Therefore, the approach provides a robust and practical tool to support climate-informed decision-making with coherent, accurate and operationally viable forecasts.

*Code and data availability.* All datasets used in the study are publicly available. SEAS5 predictions and ERA5 reanalysis data are available from the Copernicus Climate Data Store (CDS; https://cds.climate.copernicus.eu/). The multi-annual predictions, decadal predictions,

historical simulations and climate projections are available on the Earth System Grid Federation system (ESGF; https://esgf.github.io/). We acknowledge the use of the startR, s2dv, CSTools, multiApply, and ClimProjDiags R-language-based software packages, all of them available on the Comprehensive R Archive Network (CRAN; https://cran.r-project.org/). The code used during the study is available from the corresponding author on reasonable request.

*Author contributions.* CDT, MGD and FJDR designed the study. CDT carried out the analysis and wrote the first draft. PAB and MSC
downloaded and formatted the data. All authors contributed equally to the interpretation of results and writing thereafter.

*Competing interests.* The authors declare no competing interests.

*Acknowledgements.* This study has been supported by the European Union's Horizon Europe ASPECT project (grant agreement No. 101081460) and the Spanish national project BOREAS (PID2022-140673OA-I00) funded by MICIU/AEI/10.13039/501100011033 and by ERDF, EU. MGD is grateful for support by the AXA Research Fund. VT acknowledges the Beatriu de Pinós program (2022 BP 00227)
from the Secretariat of Universities and Research of the Research and Universities Department of the Generalitat de Catalunya. NPZ ac-



knowledges her AI4S fellowship within the "Generación D" initiative by Red.es, Ministerio para la Transformación Digital y de la Función Pública, for talent attraction (C005/24-ED CV1), funded by NextGenerationEU through PRTR.



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
