# Peer review of "Seamless climate information for the next months to multiple years: merging of seasonal and decadal predictions, and their comparison to multi-annual predictions"

_EGUsphere, 2025_

## Author Comment (AC1)

**Reply on RC1**

In the manuscript entitled 'Seamless climate information for the next months to multiple years: merging of seasonal and decadal predictions, and their comparison to multi-annual predictions', Delgado-Torres and colleagues evaluate the added value of seamless forecasts from multi-annual predictions and from several methods based on constraining large ensembles of simulations, in comparison with seasonal and decadal prediction systems, as well as with 'non-initialized' large ensembles of historical simulations, to predict the Niño 3.4 index and spatial fields of temperature, precipitation, and sea-level pressure. Overall, I found that the authors carried out interesting analyses that highlight the relevance of both multi-annual predictions and constraining methods, the latter being a cost-effective alternative that can be updated much more frequently. I have some minor issues and comments, especially regarding the evaluations.

We sincerely thank the reviewer for the constructive feedback and valuable suggestions. We address each comment in detail below, providing point-by-point responses.

**Title :**

I found the title not very clear. I am not sure that the term 'merging' is appropriate here, as there is no actual merging of seasonal and decadal predictions in the study, but rather a constraint of decadal predictions and historical simulations based on seasonal predictions. If you used the term 'merging' in the sense of combining different data, then 'blending' may be more suitable.

We thank the reviewer for this helpful comment. We agree that the term merging may suggest a direct fusion of seasonal and decadal predictions, whereas our study focuses on constraining or combining information to provide seamless climate information. In this sense, we think that "constraining" may be more suitable as it is used in the papers our study is based on (e.g. Mahmood et al. 2021, https://doi.org/10.1029/2021GL094915; Mahmood et al. 2022, https://doi.org/10.5194/esd-13-1437-2022; Solaraju-Murali et al. 2025, https://doi.org/10.1088/1748-9326/adfd73). Accordingly, we propose to revise the title as follows:

"Seamless climate information from months to multiple years: constraining decadal predictions with seasonal predictions and past observations, and their comparison to multi-annual predictions."

For consistency, we also propose to revise other parts of the manuscript where temporal merging was used:

- L8: replace "temporal merging" with "constraining".
- L13: replace "temporally merged" with "constrained".
- L17: replace "temporal merging" with "constraining".

- L38-41: replace "An alternative and more cost-effective approach is temporal merging, which attempts to build seamless multi-year forecasts by post-processing predictions for different timescales (e.g. seasonal and decadal forecasts) in combination. One strategy is the 40 constraining approach, which selects members from large ensembles of decadal predictions or climate projections that closely align with either recent observations or seasonal forecasts for the next months" with "An alternative and more cost-effective approach is constraining, which attempts to build seamless multi-year forecasts by post-processing predictions for different timescales (e.g. seasonal and decadal forecasts) in combination. This strategy selects members from large ensembles of decadal predictions or climate projections that closely align with either recent observations or seasonal forecasts for the next months".
- L67: replace "temporal merging" with "constraining".
- L71: replace "temporal merging" with "constraining".
- L344: replace "temporal merging" with "constraining".

**Introduction :**

l.38-41: I wouldn't describe the methods cited here as 'temporal merging' methods, since they do not merge time series (this approach is not used in these studies). Indeed, they use observations or decadal predictions to constrain large ensembles of non-initialized historical simulations. The term 'temporal merging' is more consistent with the study of Befort et al. (2022), cited on line 50, where historical simulations and decadal predictions are concatenated.

Thank you. As noted in our response to the previous comment, we agree to revise these lines.

**Data :**

l.96 : The term 'climate projection' with HIST as a reference is misleading, especially since there are not only climate projections but also historical simulations.

We thank the reviewer for this comment. We agree that referring to HIST as a climate projection may be misleading. We suggest revising the paragraph as follows to improve clarity:

"For historical simulations and climate projections (referred to as HIST for simplicity), we use the CMIP6 historical experiment extended with the SSP2-4.5 scenario (O'Neill et al., 2016), produced with 32 different climate models (resulting in a total of 264 ensemble members, Table S2). The historical experiment provides data until 2014, after which it is combined with SSP2-4.5 for the period 2015–2018."

**Method:**

Fig S1 : What does « accum » mean ?

The term "accum" was used to indicate the "accumulation period" over which member selection is performed (i.e. using one, two, three, or four months prior to the start date of the constrained forecast). To avoid confusion, we have removed the term "accum" from Figure S1 in the revised version.

l.110 : Can you provide more explanation on the « bias adjustments (correcting both the mean and variance) »

We thank the reviewer for this request. The mean and variance bias-adjustment is performed to ensure that the mean and the variance of the simulations is the same as in the reference dataset, as in Torralba et al. (2017, https://doi.org/10.1175/JAMC-D-16-0204.1). Please note that, because the post-processing is applied in a cross-validation mode (explained in L113-115), the mean and variance of the bias-corrected simulations will not be exactly identical to those of the reference dataset.

Therefore, we suggest to include the following lines and formula in the manuscript to provide more explanation on the bias-adjustment procedure:

"

The mean and variance bias-adjustment has been applied independently to each grid cell and forecast month to ensure that the mean and the variance of the simulations is the same as in the reference dataset, as in Torralba et al. (2017), following Formula 1. Please note that, because the post-processing is applied in a cross-validation mode, the mean and variance of the bias-corrected simulations will not be exactly identical to those of the reference dataset.

$$X_{corrected} = (X - X_{mean}) \cdot O_{sd} / X_{sd} + O_{mean} \qquad \text{(Formula 1)}$$

Where $X_{corrected}$ refers to the corrected simulation value, $X$ to the original simulation value, $X_{mean}$ and $X_{sd}$ to the climatology and standard deviation of the simulation, and $O_{mean}$ and $O_{sd}$ to the corresponding values in the reference dataset.
"

**Results:**

Fig S3b : It is confusing for the November initialization that the skill from DP just after initialization (dark green), which starts in January as indicated in the legend, is shown as starting at the same month (0) as the other dataset that begins in November. Shouldn't it instead start at month 2 to be consistent with the other dataset?

Thank you for this comment. We agree that it can be confusing and will start at forecast month 2. Please find below the revised Figure S3b:

[Figure]

l.217-219 : Indeed, this is not a very fair comparison with decadal predictions. It would be preferable to use the same representation as in Fig. S3B, based on the DP system initialized in November.

We understand the concern. However, our focus is on the skill that would be operationally available at forecast issuing. For a November forecast, this corresponds to DP runs initialized in the previous year, not the same November. For this reason, we believe our representation reflects the actual usable skill. However, for the comparison of the potential DP skill (if the predictions were available right after initialisation), we included such potential skill in Figure S3b (as well as in Figures S4 and S5 addressing the next comment). But, for the main text, we prefer to show the actual skill that the predictions have in an operational context, accounting for the delays between initialisation and availability. We are happy to keep discussing this point further if the reviewer feels a different representation would be more appropriate.

Fig S4 and S5 : As in my previous comment, it would be preferable to also include the DP system initialized in November for the November forecast in the Figures.

Thank you for the suggestion. We have added the DP skill curves (from November initialisations) to Figures S4 and S5 (please find such figures below), and will modify the figure captions accordingly. These values represent potential skill (since they could only be used operationally with a delay of several months), but they may provide useful context alongside the operationally available actual skill.

[Figure]

Figure S4. Same as Figure 1, but when the best members are selected only from the DP ensemble (top) or HIST ensemble (bottom) for predictions issued in May (left) and November (right). The skill for DP is also shown from the first forecast months after initialisation (i.e. from January; dark green) for comparison to the skill for DP initialised at the end of the previous year (i.e. previous January; green).

[Figure]

Figure S5. Same as Figure 2, but when the best members are selected only from the DP ensemble (top) or HIST ensemble (bottom) for predictions issued in May (left) and November (right). The skill for DP is also shown from the first forecast months after initialisation (i.e. from January; dark green) for comparison to the skill for DP initialised at the end of the previous year (i.e. previous January; green).

l.223-224 : If I understand the method correctly, the selected members from the DP ensemble are also initialized 5–7 months prior for the May forecast and 10–12 months prior for the November forecast. It would be interesting to see whether selecting members from the DP system initialized in November of the same year of the forecast could increase the skill in Fig. S4b.

We thank the reviewer for this suggestion. However, such a selection would not be operationally feasible, given the delay between model initialization and forecast availability. In addition, the member selection would have to be done in January the earliest, since this is

the month when the DP multi-model can be built. Thus, this would include another initialisation month in the analysis (in addition to May and November, which are already included). For these reasons, we prefer to focus on the skill that would be available at the actual forecast issuing time.

l. 224-227 : The fact that some methods using only HIST show such poor skill suggests that the predictor used for the constraint provides no information on the evolution of El Niño. Conversely, methods with skill comparable to SP and MP in the first forecast months appear to rely on more informative predictors. Are these best methods based solely on the Niño 3.4 index? And is there a common predictor among the worst methods as well?

We thank the reviewer for this insightful comment. We agree that the differences in skill between methods based on HIST reflect the varying degree of information contained in the predictors. While the DP ensemble carries information on the ENSO phase through initialisation, the HIST ensemble does not. As shown in Figures S5c and S5d, most of the best-performing constraining methods are based on seasonal predictions of the Niño3.4 index or on spatial fields of TOS. Specifically, for May initialisation, the three best methods are based on seasonal predictions of Niño3.4 for forecast months 1-5, 1-6, and 1-4. For November initialisation, the three best methods rely on seasonal predictions of TOS spatial fields (forecast months 1-4 over the Global and NoPolar regions, and forecast months 1-5 over the Global region). Interestingly, some methods based on PSL spatial fields also show relatively good performance, although they are much less frequent than those based on Niño3.4 or TOS.

Regarding the worst methods, we have checked which methods provide the lowest skill during the first forecast month. Most of the worst methods are based on observations of PSL or the NAO index. For instance, here are the five worst methods for each initialisation:
- May initialisation:
  - "OBS-based_3months_NAO"
  - "OBS-based_2months_NAO"
  - "OBS-based_4months_NAO"
  - "OBS-based_1months_NAO"
  - "SP-based_fmonths1-4_NAO"
- November initialisation:
  - "SP-based_fmonths1-2_NAO"
  - "SP-based_fmonths1-1_NAO"
  - "SP-based_fmonths1-3_NAO"
  - "SP-based_fmonths1-4_NAO"
  - "OBS-based_1months_psl_ACC_NAtl"

We will include these lines after L224-227: "These differences indicate that the variation in skill among HIST-based constrained ensembles largely depends on the predictive information contained in the chosen constraint. Since HIST simulations do not include information on the ENSO phase from initialisation, constraining methods that rely on ENSO-related predictors

perform better. In particular, most of the best methods are based on seasonal predictions of the Niño3.4 index or TOS spatial fields, with only a few PSL-based methods ranked among the best methods."

l. 239-240 It seems from these figures that many members are selected from two models (MIROC6 and CESM1). Do you have any thoughts on why this is the case ? Are these models better in their representation of El Niño?

Thank you for this observation. We have not specifically evaluated whether these models represent El Niño better. The higher number of selected members from MIROC6 and CESM1 is most likely explained by their larger ensemble sizes. To confirm this, we computed the percentage of selected members relative to the total ensemble size for each model, which revealed a more homogeneous distribution across models (please see figures below). The larger circles in some cases are due to the small ensemble size (e.g., 100% for an ensemble size of one when that member is selected). We will include these supplementary figures in the revised version after Figures S6 and S7.

[Figure]

Figure R1. As Figure S6, but showing the percentage of selected members.

[Figure]

Figure R2. As Figure S7, but showing the percentage of selected members.

Fig 6 : It would be helpful to clarify the choice of constraints for the different tests. For example, in panels 6d, e, f, is it based on HIST+DP? Similarly, for panels 6g, h, i, is it based on OBS or SP?

Thank you for pointing out this need for clarification. In Figure 6, each row tests the sensitivity of one factor, while keeping the others constant. For example, the first row compares results when selecting from DP+HIST, DP-only, or HIST-only ensembles (each shown in a separate boxplot), but all other constraint options (OBS-based, SP-based, TOS-based, PSL-based, etc.) are included within each boxplot. Similarly, the second row isolates the effect of using OBS vs SP as the constraint, while still including all possible selectable ensembles in each boxplot.

We propose to clarify this in the caption of Figure 6 as follows: "Each row isolates the sensitivity to one factor (ensemble composition, observational vs seasonal predictor, variable choice, etc.). Each boxplot within a row represents that specific factor, while encompassing all combinations of the other constraining choices."

Fig 6 : The legend for the fifth row is unclear and quite confusing. In the legend, you describe the mean absolute error for Nino3.4 or NAO (two scores), the spatial ACC, the spatial centered-RMSE, and the spatial uncentered-RMSE with respect to TOS or PSL (is this two scores, or four if TOS and PSL are tested for both centered- and uncentered-RMSE?).

However, only four distributions are highlighted in the figure to test the scores, with, for example, only one as the error index, which I assume corresponds to the mean absolute error — but is it for Nino3.4 or NAO? Is one missing?

We thank the reviewer for their suggestion regarding the description of the fifth row in Fig. 6. However, in this case, the fifth row is designed to test only the constraining metric (error index, ACC, RMSE, etc.), not the variable itself. The variable used is fixed depending on the metric (PSL or TOS for ACC and RMSE; NAO or Niño3.4 for the error index). We believe it will be clarified when adding the sentence proposed in the previous point.

**small correction:**

Fig 1 : It is hard to see the brown HIST line over the purple lines. For the legend, it would be more convenient for the reader to indicate the period over which the skill is calculated, so that this information is available directly in the legend.

Thank you for this suggestion. We will change the color of the HIST line to orange to make it more visible. Please see below the revised Figure 1. We will also change the colour in Figures S3, S4 and S5.

[Figure]

In addition, we will indicate the evaluation period directly in the figure caption. We will also include it in the rest of the figures. If you believe it would be more suitable in the legend, we are happy to make that adjustment.

Fig S3b : the dark green line is missing in the Figure legend below the x-axis.

We have added the corresponding item to the legend. Please find the figure in the answer to one of the previous comments (first comment on the Results section). Thank you for noticing this.

l.210: remove the tilde over the « 1 »

Thank you for pointing out this typo. We will correct it in the revised version.

Fig S6 : It would be easy for the reader to have directly _DP at the end of the models that correspond to the DP ensemble.

Thank you. We will include "_DP" in the name of the DP models in the Figures S6 and S7 (and in the new figures with the percentage of ensemble members per model shown above).

---

## Author Comment (AC2)

**Reply on RC2**

The article "Seamless climate information for the next months to multiple years: merging of seasonal and decadal predictions, and their comparison to multi-annual predictions" by Delgado-Torres and co-authors presents a new approach to combining different climate datasets in an efficient yet scientifically sound way. The topic itself is not new, but the way the authors describe it is quite convincing to me and could provide a good opportunity for climate services. I mainly have minor technical comments.

We sincerely thank the reviewer for the constructive feedback and valuable suggestions. We address each comment in detail below, providing point-by-point responses.

**Title**: I have no particular concerns about the title. However, the term "merging" is only used in the title, introduction, and conclusion, while elsewhere the key term is "constrained method/dataset." It might be worth considering a rephrasing for consistency.

Thank you for the comment. This issue was also raised by the other reviewer, and we agree that using consistent terminology improves clarity. We will therefore change the title to:

"Seamless climate information from months to multiple years: constraining decadal predictions with seasonal predictions and past observations, and their comparison to multi-annual predictions."

For consistency, we will also review the rest of the manuscript where "temporal merging" is mentioned, replacing it with "constraining".

**Data**: For seasonal prediction, only one forecast system has been used. Since seasonal predictions play an important role in the constraining, it should be discussed whether the results still hold when using a multi-model system or a different forecast system.

We thank the reviewer for the comment. Indeed, our (seasonal-based) constraining approach is based on the ECMWF SEAS5 system, chosen for its relatively long hindcast period (with retrospective predictions available from 1981) and its well-documented skill in ENSO prediction (e.g. Johnson et al., 2019, <a href="https://doi.org/10.5194/gmd-12-1087-2019">https://doi.org/10.5194/gmd-12-1087-2019</a>). Other C3S seasonal systems have often shorter hindcast periods in common archives (e.g. predictions available from 1993 in the Copernicus Climate Data Store), which would considerably reduce the evaluation period and, consequently, the robustness of the validation. Nevertheless, since the constraining methodology is model-independent, similar results are expected when using other skilful seasonal prediction systems or multi-model ensembles that adequately capture ENSO variability. Future studies could further assess the robustness of the methodology by applying it based on other seasonal forecast systems or multi-model ensembles. Therefore, we will add the following paragraph to the last section of the manuscript (Summary and Conclusions):

"This study used seasonal predictions from the SEAS5 system due to its relatively long hindcast period (with retrospective predictions from 1981 onwards) and its strong performance in ENSO forecasts (Johnson et al., 2019). Nevertheless, similar results would be expected when using other skilful seasonal systems or multi-model ensembles to perform the member selection. Therefore, future work could explore constraining methods that incorporate additional seasonal systems, as well as consider multiple variables simultaneously, potentially further improving the quality of seamless predictions."

**Method section**: This section is somewhat difficult to read due to the large number of numerical values and coordinates given in the text. As not everything should be moved to the supplements would it be possible to summarize it in a table (e.g., listing the regions) within the text?

Thank you for the comment. We agree that the clarity of the section can be improved. We will summarise the constraining options in a new Table 1 (please find it below), to be added in the Method section. In addition, we will remove the coordinates of the constraining region boxes from the main text and point to Figure S2, where these regions and their coordinates are displayed and described.

**Table 1**. Summary of constraints applied in this study. The definition of the constraining regions can be found in Figure S2.

| Parameter                            | Options                                                                                                                       |
|--------------------------------------|-------------------------------------------------------------------------------------------------------------------------------|
| Constraining variable                | TAS, PR                                                                                                                       |
| Constraining indices                 | Niño3.4, NAO                                                                                                                  |
| Constraining regions                 | Global, Global without the poles, Atlantic and Pacific Oceans, Pacific Ocean, North Atlantic Ocean (definition in Figure S2). |
| Constraining metric (variable-based) | Spatial correlation, center-RMSE, uncenter-RMSE                                                                               |
| Constraining metric (index-based)    | Mean absolute error                                                                                                           |
| Constraining period (OBS-based)      | Previous 1, 1-2, 1-3, 1-4 months                                                                                              |
| Constraining period (SP-based)       | Forecast month 1, 1-2, 1-3, 1-4, 1-5, 1-6                                                                                     |
| Selection type                       | OBS-based, SP-based                                                                                                           |
| Selectable ensemble                  | DCPP, HIST, DCPP+HIST                                                                                                         |

**Results and Discussion**: The main focus here is on Figures 1 and S4. The blue, purple, and even the red lines are hard to distinguish. Please consider choosing a better color scale, especially for the purple lines—perhaps grey? Figure 7 seems to have been prepared with a

different graphic tool, which gives it a clearer look, although the bright green line is again suboptimal.

Thank you very much for your suggestions to improve the clarity of the figures. Reviewer 1 also noted visibility issues (for example, the brown HIST line) which we will change to orange for better contrast.

We tested changing the purple lines to grey, but they were not very visible, and if made darker, they became too similar to the black MP line. After testing different options, we propose the following adjustments to enhance visibility: (1) increase the linewidth of the unconstrained SP, MP, DP, and HIST lines, and (2) decrease the linewidth of the constrained Best\_OBS and Best\_SP lines. The revised Figure 1 reflecting these changes is included below. We remain happy to further adjust the figures if the reviewers find them still unclear.

Regarding the figure preparation, all figures were produced using the ggplot2 R package, except for the maps, which were prepared using the s2dv R package.